# Exploring Use of the *Metschnikowia pulcherrima* Clade to Improve Properties of Fruit Wines

Dorota Kręgiel [1,*], Ewelina Pawlikowska [1], Hubert Antolak [2], Urszula Dziekońska-Kubczak [2] and Katarzyna Pielech-Przybylska [2]

[1] Department of Environmental Biotechnology, Łódź University of Technology, Wólczańska 171/173, 90-530 Łódź, Poland; ewelina.pawlikowska@dokt.p.lodz.pl

[2] Institute of Fermentation Technology and Microbiology, Łódź University of Technology, Wólczańska 171/173, 90-530 Łódź, Poland; hubert.antolak@p.lodz.pl (H.A.); urszula.dziekonska-kubczak@p.lodz.pl (U.D.-K.); katarzyna.pielech-przybylska@p.lodz.pl (K.P.-P.)

[*] Correspondence: dorota.kregiel@p.lodz.pl

**Abstract:** Mixed fermentation using *Saccharomyces cerevisiae* and non-*Saccharomyces* yeasts as starter cultures is well known to improve the complexity of wines and accentuate their characteristics. This study examines the use of controlled mixed fermentations with the *Metschnikowia pulcherrima* clade, *Saccharomyces cerevisiae* Tokay, and non-conventional yeasts: *Wickerhamomyces anomalus* and *Dekkera bruxellensis*. We investigated the assimilation profiles, enzyme fingerprinting, and metabolic profiles of yeast species, both individually and in mixed systems. The chemical complexity of apple wines was improved using the *M. pulcherrima* clade as co-starters. *M. pulcherrima* with *S. cerevisiae* produced a wine with a lower ethanol content, similar glycerol level, and a higher level of volatilome. However, inoculation with the *Dekkera* and *Wickerhamomyces* strains may slightly reduce this effect. The final beneficial effect of co-fermentation with *M. pulcherrima* may also depend on the type of fruit must.

**Keywords:** fruit wine; *Metschnikowia pulcherrima*; co-cultures; metabolic profiles

## 1. Introduction

Wines have become an integral part of the culture in many countries. Grape wine is one of the most widely consumed alcoholic beverages in the world. In Poland, where the summers are usually short and characterized by moderate or low temperatures, the climate is not suitable for growing grapes. However, other fruits are used for winemaking, the most popular being apples and antioxidant-rich berries, such as raspberries, strawberries, currants, or chokeberries [1]. In some Polish regions, fruit wines are recognized as traditional or regional products [2]. With increasing consumer demand, wider varieties of fruit wines are entering the market [3]. Fruit wines are prized for their refreshing tastes, which can accompany any cuisine. However, technical challenges, such as ensuring acid stability, adjusting sugar quantities, and obtaining the proper chemical characteristics, may restrict the production of fruit wines.

Fermentation is a complex biochemical process in which wine yeasts play fundamental roles in transforming sugars into ethanol, carbon dioxide, and other metabolites. The quality of wine is conditioned by several factors, including the yeast strains used [4]. The yeast strain can influence both the fermentation and conservation of wines. Over the last few decades, major advances have been made toward understanding the roles of different yeasts in the fermentation process. Currently, pure cultures of *Saccharomyces cerevisiae* or *S. bayanus* strains are mainly used in winemaking [5]. Most winemakers prefer to use commercial *Saccharomyces* sp. starters, which guarantee predictable results and reproducibility. On the other hand, the extensive use of globally distributed commercial starters leads to organoleptic 'flattening' and uniformization. In recent years, alongside conventional commercial starters, non-*Saccharomyces* yeasts have become available. Non-*Saccharomyces* yeasts

are generally unable to complete alcoholic fermentation [6,7]. However, non-*Saccharomyces* species can modulate the wine aroma profile, via esterase and glucosidase activities. They can also increase the glycerol content, lower the alcohol content, and exert proteolytic and pectinolytic activities that lead to enrichment of the aroma profile [8,9].

The fermentation process can be divided into three sequential phases based on the predominant yeast type present. *Hanseniaspora*/*Kloeckera*, *Pichia*, *Candida*, and *Metschnikowia* yeasts predominate in the initial fruit yeast phase. *Saccharomyces* sp. then dominates in the main fermentation phase. Finally, the maturation phase may be dominated by *Dekkera* and *Brettanomyces* yeasts. [10]. Of the genera active in the early fermentation phase, *Metschnikowia* sp. seems the most interesting. *Metschnikowia* sp. exerts moderate fermentation power but has interesting enzymatic activities involving aromatic and color precursors [11,12]. Phylogenetic analysis of the barcode sequence of *Metschnikowia* strains producing red pigment (pulcherrimin) has enabled the creation of a special group designated as the *M. pulcherrima* clade [13]. The antimicrobial activity of these yeasts is based on the depletion of iron in the medium. This depletion is caused by interaction with pulcherriminic acid, which is a precursor of pulcherrimin [14]. In this way, the environment becomes inhospitable to other microorganisms that require iron for their development. Pulcherrimin production stimulates effective inhibition against several non-*Saccharomyces* yeasts (*Candida* spp., *Brettanomyces/Dekkera*, *Wickerhamomyces*) and molds (*Botrytis cinerea*, *Penicillium*, *Alternaria*, etc.). It may be emphasized that *S. cerevisiae* seems unaffected by this antimicrobial activity [15]. It is also worth noting that pulcherrimin also shows other biological activities. Its unique nature makes it a good cell protectant against light and stress temperatures [16]. Therefore, in addition to having wide spectra of enzymatic activity, the *M. pulcherrima* clade is an interesting candidate for improving wine quality. However, recent studies by Mencher and co-workers showed that *M. pulcherrima* represses aerobic respiration in *S. cerevisiae*, which may suggest a direct response to cocultivation in wines [17].

In this study, we evaluate the chemical properties of Polish fruit wines fermented by yeast monocultures or in co-cultures with conventional and non-conventional yeasts. Particular attention was paid to compatibility of *M pulcherrima* with *S. cerevisiae* in producing different fruit wines, in the presence of other non-conventional yeasts.

## 2. Materials and Methods

### 2.1. Yeast Cultures

The yeast strains used in this study are presented in Table 1.

**Table 1.** Yeast strains used in the study.

| Strain | Origin | GenBank Accession Number | References |
|---|---|---|---|
| *Saccharomyces cerevisiae* Tokay (winery strain) LOCK0203 | LOCK * | - | [15] |
| *Metschnikowia pulcherrima* NCYC747 | NCYC ** | - | [15] |
| *Metschnikowia sinensis* LOCK1143 | Strawberry fruits | MK612102 | [15] |
| *Dekkera bruxellensis* NCYC D5300 | Fruit-flavored mineral water | LT908481 | [18] |
| *Wickerhamomyces anomalus* NCYC D5299 | Fruit-flavored mineral water | LT908480 | [18] |

* Collection of Pure Cultures of Industrial Microorganisms, Lodz University of Technology, Poland (LOCK); ** National Collection of Yeast Cultures; Norwich, United Kingdom (NCYC).

The yeast strain *S. cerevisiae* Tokay is widely used in Poland for wine production. *M. pulcherrima* NCYC747 was obtained from the National Collection of Yeast Cultures (UK). Other strains were the isolates from Polish fruits (*M. sinensis* LOCK1143) and contaminated soft drinks (*D. bruxellensis* NCYC D5300 and *W. anomalus* NCYC D5299). Our previous studies had shown that *S. cerevisiae* Tokay is sensitive to toxins produced by killer yeasts [19].

Previous results also point to the *M. pulcherrima* clade being harmful to both *Dekkera* and *Wickerhamomyces* strains [15]. This may be important when creating co-cultures.

The yeasts were stored at −18 °C in a microbank storage system (Microbank®, Bi-omaxima, Lublin, Poland). A single loop was transferred onto Potato Dextrose Agar (PDA) (Merck Millipore, Darmstadt, Germany) and incubated at 28 ± 2 °C for two days to activate the strains. Then, single colonies were streaked onto the PDA medium to ensure the purity of the cultures.

The yeast inoculums were prepared after incubation at 30 °C for 48 h on YPD agar. The optical density of the inoculum suspensions was measured as 6.0 degrees on the MacFarland scale (symbol °McF) using a DEN-1 densitometer (Merck Millipore, Darmstadt, Germany).

### 2.2. Assimilation Profiles

The assimilation profiles of the tested yeasts were determined using API 20 C AUX tests (bioMérieux, Lyon, France), according to the manufacturer's instructions, as described previously by [15]. The ability of the yeasts to assimilate fructose (not present in the API set) was evaluated using the conventional method for yeast identification [20].

### 2.3. Enzymatic Profiles

The enzymatic profiles of the tested yeasts were estimated using API ZYM tests (bioMerieux, Lyon, France), according to the manufacturer's recommendations. The suspensions that showed visible changes in the color of the medium were considered to demonstrate enzymatic activity. Enzyme activity was graded from 0 to 5 by comparing the developed color to the API-ZYM color reaction chart, where '0' indicates a negative test and '5' indicates a high positive test [15].

### 2.4. Fermentation Media

Apples (Golden Delicious variety) and chokeberries (Nero variety) were harvested and purchased in September 2018 from local ecological fruit orchards (Lodz, Poland; 51°46′36″ N, 19°27′17″ E). The apples and chokeberries were washed using tap water and ground in a mechanical grinder. Pectinolysis of the pulps was carried out with pectinases and arabanases using Rohapect 10 L (AKE, Pabianice, Poland) at a dose of 0.5 g/kg of fruits. Before fermentation, extracts of the apple must, and apple/chokeberry (2:1) must were standardized using sucrose and tap water to obtain 9.5 °Bx and 13.0 °Bx, respectively. The basic chemical components of the fruit musts are presented in Table 2.

**Table 2.** Basic chemical characteristics of musts obtained from fruits.

| Must | Glucose [g/L] | Fructose [g/L] | Arabinose [g/L] | Glycerol [g/L] | Ethanol [g/L] | Extract [°Bx] | pH |
|---|---|---|---|---|---|---|---|
| Apple | 168.40 ± 9.45 | 111.56 ± 7.23 | <LOD * | <LOD | <LOD | 26.82 ± 1.21 | 3.70 ± 0.15 |
| Apple/chokeberry | 174.28 ± 11.34 | 100.68 ± 8.34 | 7.72 ± 0.34 | 0.12 ± 0.03 | <LOD | 27.22 ± 3.20 | 3.13 ± 0.05 |

* LOD—Limit of detection, LOQ—Limit of quantification. The limit of detection (LOD) and the limit of quantification (LOQ) determined compounds were as follows: glucose: 0.355 g/L (LOD) and 1.077 g/L (LOQ); fructose: 0.993 g/L (LOD) and 3.008 g/L (LOQ); arabinose: 0.262 g/L (LOD) and 0.797 g/L (LOQ); glycerol: 0.041 g/L (LOD) and 0.124 g/L (LOQ); ethanol: 0.225 g/L (LOD) and 0.682 g/L (LOQ).

### 2.5. Fermentation Trials

Sterile Erlenmeyer flasks (volume 100 mL) were filled with 50 mL of the non-pasteurized musts. All samples were inoculated with 2.5 mL of yeast suspensions (5% *v/v*). The flasks were closed with fermentation airlocks and silicone stoppers to allow $CO_2$ to escape. The samples were then incubated without agitation at 30 °C. The weight loss of the flasks due to the release of $CO_2$ was monitored each day of the fermentation period (i.e., constant weight for three consecutive days). The fermentation was considered finished after stabilization of the sample weight (17 ÷ 20 days). After fermentation, the samples were centrifuged at 4 °C, 10,000× *g* for 10 min (centrifuge 5804R, Eppendorf, Germany). The supernatant was

further analyzed by chromatographic techniques. Prior to chromatography, clear liquid samples were prepared by filtration using 0.45 μm polyethersulfone membranes (Merck Millipore, Darmstadt, Germany).

### 2.6. HPLC Analysis

The profiles of the main saccharides, acetic acid, glycerol, methanol, and ethanol in the young wines were determined by HPLC (Agilent 1260 Infinity, Agilent Technologies, Santa Clara, CA, USA) with a Hi-Plex H column (7.7 × 300 mm, 8 μm; Agilent Technologies, Santa Clara, CA, USA) and a refractive index detector at 55 °C. The column temperature was maintained at 60 °C. A 5 mM solution of $H_2SO_4$ was used as a mobile phase at a flow rate of 0.7 mL/min with a sample volume of 20 μL [21]. Samples were analyzed as received and after 10-times dilution in ultrapure water. Obtained data were processed with the use of OpenLab CDS Chemstation software Rev. C.01.06 (Agilent Technologies, Santa Clara, CA, USA).

Standard solutions of pure reagents in ultrapure water were prepared to quantify the concentration of the analyzed compounds in the range of 1.5–30.0 g/L for glucose, fructose, and ethanol, 0.5–10.0 g/L for arabinose, 0.04–9.2 g/L for glycerol, and 0.02–1.25 g/L for acetic acid. The linearity of obtained calibration curves was satisfactory in the whole tested range with an $R^2$ value of at least 0.9997. The limit of detection (LOD) and limit of quantification (LOQ) were calculated according to the method proposed by Haubax and Vos [22].

### 2.7. GC-MS Analysis

To identify and quantify the volatiles in the fruit wines after fermentation, we used an Agilent 7890A GC (Agilent Technologies, Santa Clara, CA, USA) gas chromatograph equipped with an Agilent MSD 5975C quadrupole mass spectrometer and an Agilent 7697A headspace analyzer. The headspace sampler was connected to the gas chromatograph via a transfer line through the split-splitless injector. A Rxi-5ms capillary column (60 m × 0.25 mm × 0.25 μm; Restek, Bellefonte, PA, USA) was used to separate the compounds. The initial GC oven temperature was set to 30 °C and held for 6 min, then ramped up by 5 °C/min to 80 °C (held for 3 min), and again ramped up 10 °C/min to 230 °C. This final temperature was maintained for 6 min. The carrier gas was helium, with a flow rate of 1.2 mL/min. Before analysis, a 20 mL headspace vial was filled with a 7 mL sample of wine and closed tightly. Headspace conditions were as follows: the temperatures of the oven, loop, and transfer line were set at 50 °C, 60 °C, and 70 °C, respectively. The time of vial equilibration and injection durations were 20 min and 0.7 min, respectively. During sample equilibration, the vial was shaken (136 shakes/min). The temperature of the injector was 250 °C. Injections were made in split mode (10:1). The temperature of the MSD ion source, transfer line, and quadrupole was 230 °C, 250 °C, and 150 °C, respectively. The ionization energy was 70 eV.

First, qualitative analysis was performed in full scan ions monitoring mode (SCAN) to identify volatiles presented in wine samples by comparing their mass spectra with those of standard compounds and with the mass spectra of the NIST/EPA/NIH Mass Spectra Library (Version 2.0g). Next, the quantitative analysis of volatile compounds in the wine samples was performed using the external calibration method. Quantitative analysis was performed in selected ion monitoring mode (SIM).

The calibration standards were prepared by the dilution series of external analytical standards mixture. Linearity of calibration curves was tested between: 2 and 100 mg/L for ethyl acetate and acetaldehyde, 1 and 200 mg/L for 3-methylbutan-1-ol and 2-methylbutan-1-ol, 0.5 and 100 mg/L for 2-methylpropan-1-ol, 0.25 and 10 mg/L for propan-1-ol, 0.005 and 0.25 mg/L for propanal and pentanal, 0.1 and 1 mg/L for furan-2-carbaldehyde, 0.1–5 mg/L for butane-2,3-dione and 3-methylfuran, 0.005-1 mg/L for 1,1-diethoxyethane, 0.01–5 mg/L for ethyl formate, 0.01–25 mg/L for methyl acetate, 2 and 50 μg/L for pentan-2one, and 0.5–100 μg/L for other esters (ethyl propanoate, ethyl-2-methylpropanoate, 2-methylpropyl

acetate, ethyl butanoate, 2-methylbutyl acetate, 3-methylbutyl acetate, ethyl hexanoate, ethyl octanoate, ethyl decanoate, ethyl 2-methylbutanoate, and ethyl-3-methylbutanoate). The correlation coefficients of the external standards calibration curves were 0.99 on average. The values of LOD and LOQ were calculated by the method based on the standard deviation of the response and the slope of the calibration curve at levels approximating the LOD [22]. The obtained data were analyzed using Agilent MassHunter software (Agilent Technologies, Santa Clara, CA, USA).

*2.8. Statistics*

The obtained results of the quantitative analyses were presented as the mean $\pm$ SD of three separate experiments (each variant of wine was performed in three replicates and one technical sample). The performed normal distribution test (Shapiro-Wilk test) showed that the chromatographic results obtained did not have a normal distribution. Therefore, to evaluate the differences between the concentration of volatile compounds determined in the young wines obtained by using tested yeast strains, we used non-parametric Kruskal-Wallis with post hoc Dunn's test at the significance level of 0.05. Analysis was performed using XLSTAT® data analysis software (version 2022.2.1, Addinsoft, New York, NY, USA).

**3. Results and Discussion**

*3.1. Assimilation Profiles*

Fruits contain a range of naturally occurring sugars that make them taste sweet and flavourful. These sugars include disaccharides, such as sucrose, and monosaccharides such as fructose and glucose. Fruits contain various different sugars, with the ratio depending on the specific type and variety of fruit [23]. This fact stimulated us to research the assimilation profiles of the yeast strains used in our study. The tested yeasts represent different genera with various assimilation profiles. Of the sugars used to characterize the tested yeasts, glucose, fructose, and sucrose were utilized by all the tested strains. Arabinose was assimilated only by *M. sinensis* strain (Table 3).

**Table 3.** Assimilation profiles of the tested yeast strains.

| Yeast Strain | Glucose | Fructose | Sucrose | Arabinose | Cellobiose |
|:---:|:---:|:---:|:---:|:---:|:---:|
| *S. cerevisiae* | + * | + | + | − | − |
| *M. pulcherrima* | + | + | + | − | + |
| *M. sinensis* | + | + | + | + | + |
| *D. bruxellensis* | + | + | + | − | + |
| *W. anomalus* | + | + | + | − | + |

* Symbols: "+": positive assimilation test, "−" negative assimilation test.

Based on these results, the interactions between strains play a crucial role in the fermentation process. Arabinose present in apple/chokeberry must was assimilated by *M. sinensis* strain. Other assimilation profiles may also be important to produce volatiles. For example, almost all the tested strains (except *S. cerevisiae*) showed positive results for the assimilation of cellobiose, which is a known substrate of β-glucosidase and is involved in the production of monoterpenes related to the aromatic compounds in wine [24].

*3.2. Enzymatic Profiles*

Enzymatic activity plays an important role in developing wine aroma and improving the sensory properties of wine [25]. For example, hydrolysis of glycosyl-glucosides by yeast glucosidases enhances the content of aroma profiles. Numerous studies have shown that yeasts involved in vinification processes possess beta-glucosidase activity [25–27]. Arylamidases (proteases) release amino acids as precursors of aromatic compounds. Therefore, cystine arylamidase, leucine arylamidase, valine arylamidase, acid phosphatase, and naphthol-AS-BI-phosphohydrolase each have significant roles in enhancing aroma profiles during fermentation [19]. In this study, we investigated enzymatic profiles of all the tested

strains, with special attention to the activities of proteases, esterases, phosphatases, and glycosidases (Table 4).

**Table 4.** Enzymatic profiles of the tested yeast strains.

| Enzymes | | Yeast Strains | | | | |
|---|---|---|---|---|---|---|
| **Classes** | **Name** | ***S. c.*** * | ***M. p.*** | ***M. s.*** | ***D. b.*** | ***W. a.*** |
| Proteases | Leucine arylamidase | 5 | 4 | 5 | 3 | 4 |
| | Valine arylamidase | 4 | 3 | 4 | 2 | 4 |
| | Cystine arylamidase | 4 | 1 | 3 | 1 | 3 |
| Esterases | Esterase C4 | 4 | 3 | 4 | 2 | 3 |
| | Esterase C8 | 4 | 3 | 4 | 1 | 3 |
| Phosphatases | Alkaline phosphatase | 4 | 1 | 1 | 3 | 0 |
| | Acid phosphatase | 5 | 2 | 4 | 3 | 3 |
| | Naphtol-AS-BI-phosphohydrolase | 4 | 4 | 4 | 4 | 1 |
| Glycoside hydrolases | α-Glucosidase | 3 | 5 | 4 | 4 | 4 |
| | β-Glucosidase | 0 | 3 | 3 | 4 | 4 |

* Symbols: *S. c.—Saccharomyces cerevisiae, M. p.—Metschnikowia pulcherrima, M. s.—Metschnikowia sinensis, D. b.—dekkera bruxellensis. W. a.—Wickerhamomyces anomalus*. Gray-scale levels express colour intensity developed after enzyme reactions, in addition to numerical scale.

Proteases and phosphatases (leucine arylamidase, valine arylamidase, cystine arylamidase, naphthol-AS-BI-phosphohydrolase), as well as esterases were especially active in the *S. cerevisiae* and *M. sinensis* strains. The moderate activity of esterases was detected in the *W. anomalus* strain. α-Glucosidase activity was found in all the tested strains. β-glucosidase was noted at a similar level in all non-Saccharomyces yeasts. This is worth noting because β-glucosidase activity is related to the release of terpenes into wine, especially in the context of the ability of the yeasts to assimilate cellobiose [24].

In summary, the tested yeasts offer multi-enzyme pathways for synthesizing different chemicals of importance to wine complexity. The possible differentiation of yeast activity at different stages of fermentation is also important. In subsequent studies, these strains were, therefore, assessed not only in terms of the differentiation of their enzymatic activities but also for possible interactions between strains constituting the inoculum and/or other yeasts of natural microbiota of fermentation media. According to the literature, *Metschnikowia* spp. and other oxidative yeasts predominate in the initial phase. *S. cerevisiae* dominated in the main fermentation phase. The last phase may be dominated by *Dekkera/Brettanomyces* type yeasts [10]. The final effect on the chemical characteristics of wines is thus the result of different mechanisms and reactions, including between yeasts and compounds present in fruit musts.

*3.3. HPLC Analysis*

The wines fermented with mono- and co-cultures were characterized by diverse chemical profiles (Table 5).

**Table 5.** HPLC profiles of wines obtained by mono-cultures and co-cultures of yeasts.

| Wine | Strain(s) | Compound [g/L] | | | | | |
|---|---|---|---|---|---|---|---|
| | | **Glucose** | **Fructose** | **Glycerol** | **Acetic Acid** | **Methanol** | **Ethanol** |
| Apple | *S. cerevisiae* | 44.77 [b] ± 1.22 | 31.39 [b] ± 0.92 | 4.70 [ab] ± 0.22 | 0.29 [ab] ± 0.02 | 1.39 [a] ± 0.43 | 92.53 [a] ± 3.21 |
| | *M. pulcherrima* | 109.88 [a] ± 3.34 | 69.33 [ab] ± 3.22 | 0.23 [b] ± 0.12 | 0.03 [c] ± 0.01 | <LOD * | 22.84 [ab] ± 1.01 |
| | *M. sinensis* | 114.36 [a] ± 4.21 | 77.10 [ab] ± 4.02 | 1.39 [ab] ± 0.20 | 0.31 [ab] ± 0.02 | <LOD | 12.21 [b] ± 1.13 |
| | *D. bruxellensis* | 106.89 [a] ± 3.17 | 70.29 [ab] ± 3.89 | 1.29 [ab] ± 0.19 | 0.39 [ab] ± 0.02 | 0.49 [a] ± 0.11 | 29.88 [ab] ± 1.03 |
| | *W. anomalus* | 65.63 [ab] ± 1.82 | 123.37 [a] ± 1.22 | 1.62 [ab] ± 0.12 | 0.46 [a] ± 0.04 | <LOD | 18.75 [ab] ± 1.21 |
| | *S. cerevisiae + M. pulcherrima* | 58.15 [ab] ± 1.46 | 35.37 [ab] ± 2.11 | 4.79 [ab] ± 0.92 | 0.22 [b] ± 0.09 | 1.45 [a] ± 0.81 | 82.41 [ab] ± 2.41 |
| | *S. cerevisiae + M. sinensis* | 53.47 [ab] ± 2.02 | 33.17 [b] ± 0.92 | 4.14 [ab] ± 0.85 | 0.26 [ab] ± 0.06 | <LOD | 84.16 [ab] ± 1.01 |
| | *S. cerevisiae + D. bruxellensis + W. anomalus* | 64.65 [ab] ± 1.22 | 39.68 [ab] ± 2.22 | 5.16 [ab] ± 0.78 | 0.33 [ab] ± 0.09 | <LOD | 79.32 [ab] ± 1.34 |
| | *S. cerevisiae + D. bruxellensis + W. anomalus + M. pulcherrima* | 62.30 [ab] ± 3.20 | 37.10 [ab] ± 3.01 | 5.13 [ab] ± 0.91 | 0.31 [ab] ± 0.08 | <LOD | 79.76 [ab] ± 1.19 |
| | *S. cerevisiae + D. bruxellensis + W. anomalus + M. sinensis* | 60.10 [ab] ± 4.36 | 39.75 [ab] ± 2.23 | 5.33 [a] ± 0.61 | 0.37 [ab] ± 0.11 | <LOD | 81.77 [ab] ± 2.18 |
| Apple/Chokeberry | *S. cerevisiae* | 62.0 [b] ± 2.31 | 42.47 [ab] ± 1.05 | 5.08 [a] ± 0.41 | 0.23 [ab] ± 0.08 | 0.47 [ab] ± 0.11 | 77.91 [a] ± 4.26 |
| | *M. pulcherrima* | 108.4 [ab] ± 3.67 | 73.90 [ab] ± 3.27 | 1.67 [ab] ± 0.30 | 0.31 [ab] ± 0.09 | 0.09 [b] ± 0.01 | 13.32 [ab] ± 1.01 |
| | *M. sinensis* | 156.60 [a] ± 5.32 | 73.56 [ab] ± 3.33 | 1.29 [ab] ± 0.21 | 0.30 [ab] ± 0.09 | 0.20 [ab] ± 0.11 | 11.49 [b] ± 0.99 |
| | *D. bruxellensis* | 85.92 [ab] ± 2.39 | 67.46 [ab] ± 2.89 | 1.18 [b] ± 0.11 | 0.31 [ab] ± 0.01 | 0.45 [ab] ± 0.10 | 25.55 [ab] ± 1.32 |
| | *W. anomalus* | 89.74 [ab] ± 3.01 | 79.40 [a] ± 3.33 | 1.26 [ab] ± 0.36 | 0.40 [ab] ± 0.07 | 0.10 [b] ± 0.01 | 16.29 [ab] ± 1.33 |
| | *S. cerevisiae + M. pulcherrima* | 75.29 [ab] ± 1.79 | 42.84 [ab] ± 2.31 | 4.57 [ab] ± 0.30 | 0.23 [ab] ± 0.01 | 0.43 [ab] ± 0.11 | 70.78 [ab] ± 4.31 |
| | *S. cerevisiae + M. sinensis* | 73.38 [ab] ± 1.28 | 42.78 [ab] ± 1.91 | 3.85 [ab] ± 0.21 | 0.21 [b] ± 0.04 | 0.59 [a] ± 0.14 | 72.47 [ab] ± 3.26 |
| | *S. cerevisiae + D. bruxellensis + W. anomalus* | 77.29 [ab] ± 1.31 | 44.22 [ab] ± 1.65 | 4.61 [ab] ± 0.57 | 0.37 [ab] ± 0.09 | 0.35 [ab] ± 0.09 | 70.69 [ab] ± 2.99 |
| | *S. cerevisiae + D. bruxellensis + W. anomalus + M.pulcherrima* | 61.52 [b] ± 2.01 | 37.77 [b] ± 1.01 | 4.07 [ab] ± 0.91 | 0.47 [a] ± 0.09 | <LOD | 70.22 [ab] ± 2.34 |
| | *S. cerevisiae + D. bruxellensis + W. anomalus + M. sinensis* | 76.54 [ab] ± 1.35 | 44.50 [ab] ± 1.30 | 4.77 [ab] ± 0.89 | 0.38 [ab] ± 0.06 | 0.35 [ab] ± 0.08 | 70.75 [ab] ± 2.11 |

* LOD—Limit of detection, LOQ—Limit of quantification. [a–c]—mean values in the rows with common letters are not statistically different as obtained by Kruskal–Wallis test ($\alpha$-0.05) with multiple pairwise comparisons using Dunn's procedure. The limit of detection (LOD) and the limit of quantification (LOQ) determined compounds were as follows: glucose: 0.355 g/L (LOD) and 1.077 g/L (LOQ); fructose: 0.993 g/L (LOD) and 3.008 g/L (LOQ); glycerol: 0.041 g/L (LOD) and 0.124 g/L (LOQ); acetic acid: 0.007 g/L (LOD) and 0.020 g/L (LOQ); methanol: 0.028 g/L (LOD) and 0.085 g/L (LOQ); ethanol: 0.225 g/L (LOD) and 0.682 g/L (LOQ).

In young apple wine obtained with monocultures, the percentage of sugar consumption ranged from 31.61% (*M. sinensis*) to 74.22% (*S. cerevisiae*). In the case of the mixed cultures, the percentage of sugar consumption ranged from 62.73% (*S. cerevisiae + W. anomalus + D. bruxellensis*) to 69.05% (*S. cerevisiae + M. sinensis*). The higher content of fructose after fermentation with *W. anomalus* in comparison to sugar content in fruit musts may be explained by better assimilation of glucose originating from sucrose. The highest content of ethanol after fermentation was noted for wine samples with *S. cerevisiae* Tokay (92.53 g/L). The lowest content of ethanol was observed for *M. sinensis* (12.21 g/L). For *M. pulcherrima*, *W. anomalus*, and *D. bruxellensis*, the values for content of ethanol were 22.84 g/L, 18.75 g/L, and 29.88 g/L, respectively. It is worth noting that the mixed cultures of the Tokay strain with other non-Saccharomyces yeasts resulted in a lower concentration of ethanol compared to fermentation with *S. cerevisiae* Tokay as a monoculture. Of the mixed cultures, the highest ethanol concentration was noted in the wines fermented with Tokay and *M. pulcherrima* (82.41 g/L) and Tokay and *M. sinensis* (84.16 g/L). The use of mixed yeast cultures for apple juice fermentation slightly affected the formation of acetic acid and glycerol compared to the monocultures. Acetic acid content ranged from 0.03 g/L to 0.46 g/L for the monocultures and from 0.22 g/L to 0.37 g/L for the co-cultures of yeasts. Both the acetic acid content (0.03 g/L) and the glycerol content (0.23 g/L) were the lowest for the *M. pulcherrima* strain, while the highest glycerol concentration was detected for Tokay (4.7 g/L). However, for co-cultures of Tokay and *M. sinensis* this value was only slightly lower, at 4.14 g/L.

In apple/chokeberry young wine, the lowest saccharide consumption was found for samples obtained with *Metschnikowia sinensis* (16.29%). At the same time, the highest was noted for the samples after fermentation with mixed populations of *S. cerevisiae*, *M. pulcherrima*, *W. anomalus*, and *D. bruxellensis* (63.89%). This was slightly higher than in the case of fermentation with the Tokay strain alone (61.90%). Ethanol concentration ranged from 11.49 g/L (*M. sinensis*) to 77.91 g/L (Tokay). Analogously to the fermentation of apple juice, lower concentrations of ethyl alcohol were noted for the fermentation of apple-chokeberry by mixed cultures. These concentrations ranged from 70.22 g/L (*S. cerevisiae + M. pulcherrima + W. anomalus + D. bruxellensis*) to 72.42 g/L (*S. cerevisiae* and *M. sinensis*). Lower concentrations of glycerol (3.85 g/L ÷ 4.77 g/L) were found for wines after fermentation by mixed cultures compared to *S. cerevisiae* Tokay (5.08 g/L). Acetic acid content ranged from 0.21 g/L (Tokay and *M. sinensis*) to 0.47 g/L (*S. cerevisiae + M. pulcherrima + W. anomalus + D. bruxellensis*).

All tested non-Saccharomyces yeasts were characterized by much lower ethanol production. This has been widely described in the literature [28,29]. In the past, non-*Saccharomyces* species have been considered poor fermenters because of their low fermentative efficiency, low tolerance to enological additives such as sulfur dioxide, and production of acetic acid. However, recent findings have encouraged researchers and wine producers to reconsider the ability of non-*Saccharomyces* species to work in synergy with *S. cerevisiae* to produce wine with high sensory quality and low alcohol levels [11,30–32]. In the present study, *Metschnikowia* species, in combination with *S. cerevisiae* and other yeasts, were found to decrease the synthesis of ethanol. Acetic acid formation also decreased in the presence of *Metschnikowia* spp. Similar results were reported by Hranilovic and co-workers for grape wines [32]. This may indicate the positive influence of the *M. pulcherrima* clade on the aroma and taste quality of wine obtained by co-culture with *S. cerevisiae*.

Glycerol is one of the main ingredients responsible for the mouth-feel characteristics of wine [33]. In apple wine, the glycerol value in mixed cultures of *S. cerevisiae* and *Metschnikowia* spp. was higher than in the case of the Tokay monoculture. These results are similar to those obtained by Seguinot et al. and Liu et al. for grape wine, where the glycerol values increased with mixed cultures [34,35]. In chokeberry/apple wine, the glycerol content in mixed cultures was slightly lower in comparison to wine with the monoculture *S. cerevisiae* Tokay.

### 3.4. GC-MS Analysis

Conventional and non-conventional yeasts have different oenological characteristics [36]. Their diverse secondary metabolic pathways and enzymatic profiles (esterases, β-glycosidases, lipases, and proteases) contribute to the increased diversity of flavor phenotypes in wines. The concept of flavor phenotypes is interesting for yeast selection. More than 1300 volatile compounds can now be determined in wine [37]. Our analysis of the main volatile compounds using GC-MS allowed for identifying the main components of volatilomes with concentrations above 0.0001 mg/L (Table 6). The volatilomes also depended on the type of fruit material used for the fermentation trials.

The greatest variation of volatiles was observed in the apple/chokeberry wine. Of the 26 major volatile compounds found, the lowest number was obtained from the wine fermented with *M. sinensis* (9 compounds). The highest number was for *S. cerevisie* Tokay (22 compounds) and co-cultures (17 ÷ 19 compounds). In turn, the wine from *W. anomalus* showed 10 compounds, and the wine from *M. pulcherrima* showed 12 compounds. However, despite having a greater variety of volatiles, apple/chokeberry wine fermented with the tested strains showed lower overall concentrations of volatiles. Of the 23 main volatile compounds identified for apple wine, the lowest numbers were noted for *W. anomalus* (10 compounds), *M. pulcherrima* (12 compounds), and *M. sinensis* (13 compounds). Like apple/chockeberry wine, the highest numbers of volatiles were for the monoculture of *S. cerevisie* and co-cultures with this strain (18 ÷ 19 compounds).

The lowest values of volatiles were noted for the *D. bruxellensis* strain: 50.57 mg/L in apple/chokeberry wine and 71.20 mg/L in apple wine. The strains of *Metschnikowia* spp. used as monocultures were characterized by relatively high values of total volatile compounds: 82.88 mg/L for *M. pulcherrima* and 140.59 mg/L for *M. sinensis* in apple/chokeberry wine. The highest amounts of volatiles in the young wines were found for the co-culture *S. cerevisiae* + *M. pulcherrima* + *W. anomalus* + *D. bruxellensis*: 329.47 mg/L for apple wine and 273.30 mg/L for apple/chokeberry wine. It should be emphasized that the production of volatiles in wines was correlated with the levels of enzymatic activity by β-glucosidase, esterases, and proteases of *Metschnikowia* spp. Maturano and co-workers [38] reported similar results for grape musts fermented by non-conventional yeasts.

In the apple wine, the highest concentrations of the main volatiles were of 3-methylbutan-1-ol (164.58 mg/L), produced by a co-culture *S. cerevisiae* + *M. pulcherrima* + *W. anomalus* + *D. bruxellensis* and 2-methylpropan-1-ol (66.38 mg/L) produced by a co-culture of *S. cerevisiae* + *M. sinensis*. These aliphatic higher alcohols are the main alcohols present in wines and contribute desirable complexity to wine aroma in moderate concentrations below 300 mg/L [39]. Higher concentrations of ethyl acetate (88.94 mg/L) formed in the wine fermented with a monoculture of *W. anomalus*, and acetaldehyde (46.33 mg/L) was produced by the monoculture of *M. pulcherrima*. These compounds are the components of the volatile composition that most significantly determine wine aroma [40]. Contrary to the *S. cerevisiae* Tokay strain, neither *M. pulcherrima* nor *M. sinensis* produced the following volatiles: ethyl 2-methylpropanoate, pentanal, 3-methylbutyl acetate, ethyl hexanoate, ethyl octanoate, and ethyl decanoate. Butane-2,3-dione (diacetyl) was found in the apple wine fermented with a monoculture of *M. pulcherrima*, while 3-methylfuran was found only in the samples with *M. sinensis*. These volatiles were not present in wine fermented with *S. cerevisie*. Therefore, even within the *M. pulcherrima* clade, there may be significant diversity of aromatic profiles. In wine samples fermented with a monoculture of *D. bruxellensis*, ethyl 2-methylbutanoate (0.014 mg/L) and ethyl 3-methylbutanoate (0.007 mg/L) were found. Interestingly, in samples fermented with mixed populations containing *D. bruxellensis*, only ethyl 2-methylbutanoate was identified, in concentrations from 0.005 mg/L to 0.024 mg/L. Thus, mixed populations strongly altered the aromatic profiles of the fruit wines compared to the corresponding monocultures. Similar results were reported by Antoce and Cojocaru [41] for grape wines.

**Table 6.** Volatilomes of fruit wines obtained from mono- and co-cultures of yeasts.

| Wine | | Compound (IUPAC Name) [mg/L] | Strain(s) | | | | | | | | | |
|---|---|---|---|---|---|---|---|---|---|---|---|---|
| | | | *S. c.* * | *M. p.* | *M. s.* | *D. b.* | *W. a.* | *S. c. + M. p.* | *S. c. + M. s.* | *S. c. + D. b. + W. a* | *S. c. + D. b. + W. a + M. p.* | *S. c. + D. b. + W. a + M. s.* |
| Apple | Esters | Ethyl formate | 0.149 ab ± 0.005 | 0.240 a ± 0.012 | 0.088 ab ± 0.000 | <LOD | 0.078 ab ± 0.012 | 0.159 ab ± 0.012 | 0.177 ab ± 0.014 | 0.061 b ± 0.011 | 0.127 ab ± 0.075 | 0.106 ab ± 0.025 |
| | | Methyl acetate | 0.043 bc ± 0.003 | 0.063 abc ± 0.002 | 0.154 ab ± 0.015 | <LOD | 0.526 a ± 0.011 | 0.101 abc ± 0.075 | 0.070 abc ± 0.035 | 0.033 c ± 0.009 | 0.051 abc ± 0.003 | 0.071 abc ± 0.013 |
| | | Ethyl acetate | 13.070 b ± 0.780 | 52.870 ab ± 1.022 | 42.634 ab ± 1.102 | 33.977 ab ± 0.675 | 88.941 a ± 2.011 | 24.447 ab ± 1.001 | 20.063 b ± 0.875 | 58.098 ab ± 1.003 | 56.295 ab ± 0.785 | 63.105 ab ± 3.105 |
| | | Ethyl propanoate | 0.031 ab ± 0.005 | 0.066 a ± 0.011 | 0.047 ab ± 0.003 | 0.025 ab ± 0.002 | <LOD | 0.024 ab ± 0.000 | 0.036 ab ± 0.004 | 0.022 b ± 0.003 | 0.067 a ± 0.015 | 0.061 ab ± 0.005 |
| | | Ethyl-2-methyl-propanoate | 0.004 b ± 0.000 | <LOD | <LOD | 0.084 a ± 0.012 | <LOD | 0.003 b ± 0.000 | 0.007 b ± 0.000 | 0.021 ab ± 0.002 | 0.027 ab ± 0.004 | 0.066 a ± 0.003 |
| | | 2-Methylpropyl acetate | 0.007 ab ± 0.001 | <LOD | 0.005 b ± 0.000 | <LOD | <LOD | 0.007 ab ± 0.001 | 0.006 ab ± 0.000 | 0.012 ab ± 0.000 | 0.011 ab ± 0.002 | 0.015 a ± 0.004 |
| | | Ethyl butanoate | 0.007 ab ± 0.000 | <LOD | 0.004 b ± 0.001 | 0.009 ab ± 0.000 | <LOD | 0.012 ab ± 0.004 | 0.005 b ± 0.000 | 0.014 ab ± 0.007 | 0.020 ab ± 0.002 | 0.034 a ± 0.004 |
| | | 3-Methylbutyl acetate | 0.046 ab ± 0.015 | <LOD | <LOD | 0.009 b ± 0.001 | <LOD | 0.124 a ± 0.075 | 0.087 ab ± 0.012 | 0.063 ab ± 0.011 | 0.064 ab ± 0.003 | 0.088 ab ± 0.012 |
| | | Ethyl hexanoate | 0.005 b ± 0.000 | <LOD | <LOD | 0.019 ab ± 0.003 | <LOD | 0.008 ab ± 0.000 | 0.006 b ± 0.000 | 0.008 ab ± 0.002 | 0.016 ab ± 0.007 | 0.049 a ± 0.003 |
| | | Ethyl octanoate | 0.005 b ± 0.000 | <LOD | <LOD | 0.011 ab ± 0.002 | <LOD | 0.006 ab ± 0.001 | 0.006 ab ± 0.000 | 0.006 ab ± 0.001 | 0.009 ab ± 0.002 | 0.062 ab ± 0.006 |
| | | Ethyl decanoate | 0.004 ab ± 0.000 | <LOD | <LOD | <LOD | <LOD | 0.002 b ± 0.000 | 0.005 a ± 0.001 | <LOD | <LOD | 0.002 ab ± 0.000 |
| | | Ethyl 2-methylbutanoate | <LOD | <LOD | <LOD | 0.014 ab ± 0.003 | <LOD | <LOD | <LOD | 0.005 b ± 0.000 | 0.006 ab ± 0.000 | 0.024 a ± 0.003 |
| | | Ethyl-3-methylbutanoate | <LOD | <LOD | <LOD | 0.007 ± 0.000 | <LOD | <LOD | <LOD | <LOD | <LOD | <LOD |
| | Alcohols | Propan-1-ol | 3.504 ab ± 0.125 | 1.434 b ± 0.115 | 0.913 b ± 0.056 | <LOD | 3.237 ab ± 0.012 | 3.696 ab ± 0.015 | 4.793 a ± 0.673 | 2.896 ab ± 0.912 | 2.996 ab ± 0.114 | 2.105 ab ± 0.095 |
| | | 2-Methylpropan-1-ol | 50.675 ab ± 1.105 | 52.858 ab ± 0.895 | 52.845 ab ± 0.005 | 4.956 b ± 0.235 | 3.802 b ± 0.075 | 56.185 ab ± 1.002 | 66.382 a ± 2.000 | 54.067 ab ± 1.075 | 53.008 ab ± 1.346 | 52.991 ab ± 0.895 |
| | | 3-Methylbutan-1-ol | 133.063 ab ± 2.124 | 43.286 ab ± 1.001 | 29.285 ab ± 0.997 | 12.338 ab ± 0.095 | 10.092 b ± 0.789 | 159.451 a ± 0.005 | 137.647 ab ± 3.005 | 156.879 a ± 2.015 | 164.577 a ± 3.045 | 155.604 a ± 2.974 |
| | | 2-Methylbutan-1-ol | 26.576 ab ± 0.805 | 8.213 ab ± 0.125 | 5.735 ab ± 0.764 | 1.462 b ± 0.125 | 4.429 ab ± 0.712 | 32.967 a ± 2.195 | 32.691 a ± 1.025 | 28.534 ab ± 3.025 | 29.723 ab ± 0.985 | 28.045 ab ± 1.113 |
| | Aldehydes | Acetaldehyde | 44.902 a ± 0.998 | 46.327 a ± 1.005 | 18.949 b ± 0.789 | 18.241 b ± 0.915 | 25.207 ab ± 0.985 | 36.287 ab ± 1.112 | 46.129 a ± 2.012 | 20.913 ab ± 1.002 | 22.350 ab ± 0.965 | 23.341 ab ± 1.002 |
| | | Propanal | 0.027 b ± 0.005 | 0.060 ab ± 0.002 | 0.158 a ± 0.023 | 0.047 ab ± 0.015 | 0.067 ab ± 0.011 | 0.040 ab ± 0.006 | 0.041 ab ± 0.012 | 0.059 ab ± 0.006 | 0.048 ab ± 0.001 | 0.064 ab ± 0.004 |
| | | Pentanal | 0.016 b ± 0.000 | <LOD | <LOD | <LOD | 0.021 ab ± 0.005 | 0.021 ab ± 0.002 | 0.024 a ± 0.005 | <LOD | 0.020 ab ± 0.002 | <LOD |
| | Others | 1,1-Diethoxyethane | 0.434 a ± 0.013 | 0.035 b ± 0.000 | <LOD | <LOD | <LOD | 0.255 ab ± 0.011 | 0.403 a ± 0.095 | 0.068 ab ± 0.012 | 0.060 ab ± 0.003 | 0.077 ab ± 0.004 |
| | | Butane-2,3-dione | <LOD | 0.701 ± 0.075 | <LOD | <LOD | <LOD | <LOD | <LOD | <LOD | <LOD | <LOD |
| | | 3-Methylfuran | <LOD | <LOD | 1.658 ± 0.124 | <LOD | <LOD | <LOD | <LOD | <LOD | <LOD | <LOD |
| | | Total (compounds number/amount) | 19/272.568 | 12/206.153 | 13/152.474 | 14/71.200 | 10/136.400 | 19/313.794 | 19/308.579 | 18/321.760 | 19/329.474 | 19/325.913 |

**Table 6.** *Cont.*

| Wine | | Compound (IUPAC Name) [mg/L] | Strain(s) | | | | | | | | | |
|---|---|---|---|---|---|---|---|---|---|---|---|---|
| | | | *S. c.* * | *M. p.* | *M. s.* | *D. b.* | *W. a.* | *S. c. + M. p.* | *S. c. + M. s.* | *S. c. + D. b. + W. a* | *S. c. + D. b. + W. a + M. p.* | *S. c. + D. b. + W. a + M. s.* |
| Apple/Chokeberry | Esters | Ethyl formate | 0.123 ab ± 0.040 | 0.095 b ± 0.011 | <LOD | <LOD | 2.322 a ± 0.056 | 0.208 ab ± 0.020 | <LOD | <LOD | <LOD | <LOD |
| | | Methyl acetate | 0.039 b ± 0.011 | 0.082 b ± 0.008 | 0.717 ab ± 0.017 | 18.886 a ± 1.120 | <LOD | <LOD | 0.378 ab ± 0.018 | 0.214 ab ± 0.017 | 0.320 ab ± 0.026 | <LOD |
| | | Ethyl acetate | 20.522 ab ± 0.896 | 15.396 b ± 0.876 | 39.391 ab ± 1.200 | <LOD | 85.961 a ± 2.235 | 17.622 ab ± 0.798 | 19.568 ab ± 0.865 | 52.022 ab ± 4.173 | 73.327 ab ± 5.881 | 53.092 ab ± 4.258 |
| | | Ethyl propanoate | 0.038 ab ± 0.009 | <LOD | <LOD | 0.035 ab ± 0.003 | 0.010 b ± 0.000 | 0.028 ab ± 0.009 | 0.016 ab ± 0.007 | 0.011 b ± 0.001 | 0.061 a ± 0.005 | 0.014 ab ± 0.001 |
| | | Ethyl-2-methylpropanoate | 0.009 b ± 0.003 | <LOD | <LOD | 0.134 ab ± 0.012 | <LOD | 0.004 b ± 0.001 | <LOD | 0.510 a ± 0.041 | 0.028 ab ± 0.02 | 0.457 a ± 0.037 |
| | | 2-Methylpropyl acetate | 0.006 ± 0.000 | <LOD | <LOD | <LOD | <LOD | <LOD | <LOD | <LOD | <LOD | <LOD |
| | | Ethyl butanoate | 0.012 a ± 0.001 | 0.003 b ± 0.000 | <LOD | <LOD | <LOD | <LOD | <LOD | <LOD | <LOD | <LOD |
| | | 2-Methylbutyl acetate | 0.012 a ± 0.004 | <LOD | <LOD | <LOD | <LOD | 0.020 a ± 0.003 | 0.013 a ± 0.001 | 0.020 a ± 0.007 | 0.019 a ± 0.003 | 0.018 a ± 0.004 |
| | | 3-Methylbutyl acetate | 0.133 a ± 0.010 | <LOD | <LOD | <LOD | <LOD | 0.127 a ± 0.012 | 0.136 a ± 0.023 | 0.068 a ± 0.011 | 0.083 a ± 0.012 | 0.064 a ± 0.009 |
| | | Ethyl hexanoate | 0.016 ab ± 0.004 | <LOD | <LOD | 0.066 a ± 0.009 | <LOD | 0.016 ab ± 0.003 | 0.017 ab ± 0.009 | 0.019 ab ± 0.006 | 0.016 ab ± 0.005 | 0.012 b ± 0.003 |
| | | Ethyl octanoate | 0.019 ab ± 0.009 | <LOD | <LOD | 0.032 a ± 0.011 | <LOD | 0.014 ab ± 0.002 | 0.025 ab ± 0.008 | 0.017 ab ± 0.003 | 0.024 ab ± 0.006 | 0.008 b ± 0.001 |
| | | Ethyl decanoate | 0.014 a ± 0.001 | <LOD | <LOD | 0.001 b ± 0.000 | <LOD | 0.003 ab ± 0.000 | 0.007 ab ± 0.002 | <LOD | <LOD | <LOD |
| | | Ethyl 2-methylbutanoate | <LOD | <LOD | <LOD | 0.060 a ± 0.009 | <LOD | <LOD | <LOD | 0.009 b ± 0.002 | 0.006 b ± 0.001 | 0.006 b ± 0.001 |
| | | Ethyl-3-methylbutanoate | <LOD | <LOD | <LOD | 0.029 ± 0.008 | <LOD | <LOD | <LOD | <LOD | <LOD | <LOD |
| | Alcohols | Propan-1-ol | 3.024 ab ± 0.076 | 0.763 b ± 0.089 | 2.470 ab ± 0.745 | 0.984 b ± 0.045 | 5.744 ab ± 0.843 | 5.056 ab ± 0.943 | 6.905 a ± 0.278 | 5.247 ab ± 0.313 | 4.869 ab ± 0.075 | 4.876 ab ± 0.098 |
| | | 2-Methylpropan-1-ol | 56.285 a ± 1.016 | 28.389 ab ± 0.987 | 38.909 ab ± 1.013 | 2.864 b ± 0.079 | 2.414 b ± 0.092 | 49.527 a ± 1.750 | 58.786 a ± 1.987 | 46.301 a ± 0.982 | 47.129 a ± 1.005 | 49.880 a ± 1.003 |
| | | 3-Methylbutan-1-ol | 138.955 a ± 3.065 | 18.720 b ± 1.005 | 36.057 b ± 2.019 | 8.799 b ± 0.094 | 6.809 b ± 0.123 | 113.018 a ± 4.002 | 112.34 a ± 3.128 | 23.491 b ± 0.876 | 103.272 a ± 2.978 | 22.623 b ± 0.783 |
| | | 2-Methylbutan-1-ol | 22.699 a ± 0.987 | 2.309 b ± 0.090 | 4.693 b ± 0.167 | 1.362 b ± 0.078 | 2.344 b ± 0.090 | 22.278 a ± 1.011 | 22.178 a ± 0.904 | 31.600 a ± 1.007 | 17.432 a ± 0.798 | 29.617 a ± 0.912 |
| | Aldehydes | Acetaldehyde | 20.444 abc ± 1.001 | 14.840 bc ± 0.876 | 17.639 abc ± 1.011 | 11.501 c ± 1.009 | 19.943 abc ± 0.995 | 19.677 abc ± 0.762 | 28.374 a ± 1.017 | 19.304 abc ± 0.680 | 25.352 ab ± 0.987 | 22.842 abc ± 0.607 |
| | | Propanal | 0.051 ab ± 0.009 | 0.100 ab ± 0.012 | 0.093 ab ± 0.023 | 0.132 a ± 0.067 | 0.053 ab ± 0.009 | 0.048 ab ± 0.007 | <LOD | <LOD | <LOD | 0.033 b ± 0.011 |
| | | Pentanal | 0.031 a ± 0.005 | <LOD | <LOD | <LOD | 0.013 ab ± 0.002 | 0.019 ab ± 0.001 | 0.021 ab ± 0.009 | 0.012 b ± 0.005 | 0.015 ab ± 0.004 | 0.011 b ± 0.001 |
| | | Furan-2-carbaldehyde | 0.729 a ± 0.067 | 0.413 ab ± 0.067 | <LOD | 0.826 a ± 0.076 | <LOD | 0.399 b ± 0.012 | <LOD | 0.608 a ± 0.032 | 0.579 a ± 0.067 | 0.441 ab ± 0.089 |

**Table 6.** *Cont.*

| Wine | Compound (IUPAC Name) [mg/L] | | Strain(s) | | | | | | | | | |
|---|---|---|---|---|---|---|---|---|---|---|---|---|
| | | | *S. c.* * | *M. p.* | *M. s.* | *D. b.* | *W. a.* | *S. c. + M. p.* | *S. c. + M. s.* | *S. c. + D. b. + W. a* | *S. c. + D. b. + W. a + M. p.* | *S. c. + D. b. + W. a + M. s.* |
| Apple/Chokeberry | Others | 1,1-Diethoxyethane | 0.113 [a] ± 0.034 | <LOD | <LOD | <LOD | <LOD | 0.078 [ab] ± 0.011 | 0.108 [a] ± 0.011 | 0.055 [b] ± 0.008 | 0.114 [a] ± 0.009 | 0.068 [ab] ± 0.012 |
| | | Butane-2,3-dione | 0.954 [ab] ± 0.067 | 1.768 [a] ± 0.109 | 0.619 [b] ± 0.076 | 0.358 [c] ± 0.046 | <LOD | 0.973 [ab] ± 0.078 | 1.276 [a] ± 0.101 | 0.646 [b] ± 0.076 | 0.610 [b] ± 0.045 | 0.593 [bc] ± 0.097 |
| | | 3-Methylfuran | <LOD | <LOD | <LOD | 4.500 ± 0.870 | <LOD | <LOD | <LOD | <LOD | <LOD | <LOD |
| | | Pentan-2-one | <LOD | <LOD | <LOD | <LOD | <LOD | <LOD | 0.037 [a] ± 0.012 | <LOD | 0.016 [b] ± 0.006 | <LOD |
| | Total (compounds number/amount) | | 22/264.228 | 12/82.878 | 9/140.587 | 17/50.570 | 10/125.614 | 19/229.122 | 17/250.202 | 18/180.170 | 19/273.300 | 18/184.673 |

* *S. c.*—*Saccharomyces cerevisiae*, *M. p.*—*Metschnikowia pulcherrima*, *M. s.*—*Metschnikowia sinensis*, *D. b.*—*Dekkera bruxellensis*. *W. a.*—*Wickerhamomyces anomalus*. [a–c]—mean values in the rows with common letters are not statistically different as obtained by Kruskal–Wallis test ($\alpha$-0.05) with multiple pairwise comparisons using Dunn's procedure. The limit of detection (LOD) and the limit of quantification (LOQ) determined compounds were as follows: ethyl formate: 10.3 µg/L (LOD) and 31.2 µg/L (LOQ); methyl acetate: 5.8 µg/L (LOD) and 17.6 µg/L (LOQ); ethyl acetate: 3.331 mg/L (LOD) and 10.092 mg/L (LOQ); ethyl propanoate: 1.7 µg/L (LOD) and 5.1 µg/L (LOQ); ethyl-2-methylpropanoate: 0.6 µg/L (LOD) and 1.7 µg/L (LOQ); 2-methylpropyl acetate: 0.6 µg/L (LOD) and 1.9 µg/L (LOQ); ethyl butanoate: 0.4 µg/L (LOD) and 1.1 µg/L (LOQ); 2-methylbutyl acetate: 2.4 µg/L (LOD) and 7.2 µg/L (LOQ); 3-methylbutyl acetate: 2.0 µg/L (LOD) and 6.1 µg/L (LOQ); ethyl hexanoate: 1.1 µg/L (LOD) and 3.4 µg/L (LOQ); ethyl octanoate: 0.4 µg/L (LOD) and 1.1 µg/L (LOQ); ethyl decanoate: 0.4 µg/L (LOD) and 1.1 µg/L (LOQ); ethyl 2-methylbutanoate: 0.4 µg/L (LOD) and 1.1 µg/L (LOQ); ethyl-3-methylbutanoate: 0.6 µg/L (LOD) and 1.7 µg/L (LOQ); propan-1-ol: 0.140 mg/L (LOD) and 0.423 mg/L (LOQ); 2-methylpropan-1-ol: 0.595 mg/L (LOD) and 1.802 mg/L (LOQ); 3-methylbutan-1-ol: 1.573 mg/L (LOD) and 4.767 mg/L (LOQ); 2-methylbutan-1-ol: 0.286 mg/L (LOD) and 0.866 mg/L (LOQ); acetaldehyde: 2.703 mg/L (LOD) and 8.189 mg/L (LOQ); propanal: 7.6 µg/L (LOD) and 23.1 µg/L (LOQ); pentanal: 2.7 µg/L (LOD) and 8.2 µg/L (LOQ); furan-2-carbaldehyde: 0.076 mg/L (LOD) and 0.231 mg/L (LOQ); 1,1-diethoxyethane: 9.0 µg/L (LOD) and 27.4 µg/L (LOQ); butane-2,3-dione: 0.097 mg/L (LOD) and 0.295 mg/L (LOQ); 3-methylfuran: 0.338 mg/L (LOD) and 1.025 mg/L (LOQ); pentan-2-one: 3.1 µg/L (LOD) and 9.4 µg/L (LOQ).

In apple/chokeberry wines, 3-methylbutan-1-ol (138.96 mg/L for the *S. cerevisiae* monoculture), ethyl acetate (85.96 mg/L for the *W. anomalus* monoculture), 2-methylpropan-1-ol (58.79 mg/L, co-culture of *S. cerevisiae* + *M. sinensis*), and acetaldehyde (28.37 mg/L, co-culture of *S. cerevisiae* + *M. sinensis*) were determined. Much like in the apple wines, 3-methylfuran (4.5 mg/L), ethyl 2-methylbutanoate (0.06 mg/L), and ethyl 3-methylbutanoate (0.029 mg/L) were found in the apple/chokeberry wine fermented with a monoculture of *D. bruxellensis*. However, in wines fermented with mixed populations containing *D. bruxellensis*, only ethyl 2-methylbutanoate (0.006 ÷ 0.009 mg/L) was detected.

Differences in the concentrations of higher alcohols depended on the yeast strain used as a monoculture for both apple and apple/chokeberry wines. The major higher alcohols found in the alcoholic beverages were propan-1-ol (n-propyl alcohol), iso-butanol (2-methylpropan-1-ol), and isoamyl alcohol (3-methyl-1-butanol). The concentrations of these compounds were higher in the apple wines fermented with *S. cerevisiae* Tokay compared to the samples fermented with non-Saccharomyces yeasts, by, on average, 51% for *M. pulcherrima*, 58% for *M. sinensis*, 90% for *D. bruxellensis*, and 91% for *W. anomalus*. In the case of apple/chokeberry wines, the concentrations of these compounds were higher by 77%, 79%, 94%, and 92%, respectively. However, this effect was reduced in yeast co-cultures. The concentration of higher alcohols was higher in the case of apple wines from mixed cultures. It is worth noting that a concentration of higher alcohol in the range of 300 ÷ 400 mg/L is acceptable, but concentrations below 300 mg/L give a desirable, pleasant character [42]. All the tested samples were within the limits of organoleptic acceptability.

Esters are a broad group of by-products found in fermented beverages. They can form due to the chemical condensation of carboxylic acids and alcohols. However, they are mainly products of yeast metabolism. The enzymatic synthesis of esters is catalysed by esterases and lipases, including acetyltransferases [43]. Of the esters identified in the tested samples, ethyl acetate was found in the highest quantities. *W. anomalus* was able to produce especially large amounts of this ester. The concentration of ethyl acetate was 88.9 mg/L in apple wine and 86 mg/L in apple/chokeberry wine. According to the literature, concentrations of ethyl acetate between 50 mg/L and 80 mg/L may increase the fruity sensory properties (pear and banana) of fruit wines [5,44]. In our research, these values were slightly exceeded, which may cause unpleasant taste sensations. However, the addition of other yeast cultures reduced the production of this compound to within the preferred limit. The levels of fatty acid ethyl esters did not exceed 150–160 mg/L, even in the wines obtained from monocultures. Exceeding this level can result in an undesirable odor of 'nail polish remover' or 'solvents' [31].

The chemical character of the fruit wines was found to result from the various assimilatory and enzymatic activities of the yeasts. They were also determined by the interactions between the yeasts and other microorganisms present in various states of metabolic activity in the fermentation matrix. According to the literature, yeasts other than *Saccharomyces* sp. can affect fermentation efficiency positively, neutrally, or negatively [45]. Positive interactions can result from metabolite exchange, reorientation of carbon fluxes, and modification of the NAD + /NADH balance. Acetaldehyde production, the release of beneficial products (e.g., amino acids), cell contact-dependent effects, and the production of putative quorum sensing molecules (e.g., aromatic alcohols) can also have positive effects. Negative interactions may include substrate uptake (e.g., nitrogen, glucose, oxygen), space occupation, iron sequestration, and the production of various lethal compounds [46]. It is difficult to classify the interactions between yeasts in wines, even in the case of potential biocontrol agents, because the fruit matrix can have a strong influence [47,48].

The aromatic complexity of apple wines was improved by using the *M. pulcherrima* clade as a co-starter. When co-inoculated with *S. cerevisiae*, the *M. pulcherrima* clade produced a wine with a lower ethanol content, similar glycerol level, and higher concentration of volatiles. However, inoculation with other *Dekkera* and *Wickerhamomyces* strains may reduce this effect. The apple/chokeberry matrix was a more complicated matrix for observing the fermentative activity of the *M. pulcherrima* clade. Among other substances, the

berries of *A. melanocarpa* contain anthocyanins and procyanidins with strong antioxidative and antimicrobial potential. Antimicrobial activity tests show that proanthocyanidins are the most potent antimicrobial agents in chokeberries [49]. Compounds contained in chokeberries may affect the fermentation activity and metabolism of yeasts. For example, in a study by Cakar and co-workers, chokeberry wine samples exhibited higher α-glucosidase inhibitory activity. The most active inhibitor was chlorogenic acid [50]. In our study, the activity of this enzyme was observed in all strains forming co-cultures.

This preliminary research is a first step toward evaluating fruit wines obtained using the *M. pulcherrima* clade and other nonconventional yeasts. The experiments were conducted on a small laboratory scale with a small number of samples. Therefore, we decided to apply simple statistical methods for data analysis. In general, no significant differences were found between the wine samples. Other studies in the literature have also used statistical and data mining techniques to classify wines by their characteristics, although with different feature selection frameworks and much larger numbers of wine samples and duplicates [51–54]. In future research, we will also consider more varieties of fruit wines and larger-scale wine production, as well as fruits of different origins and degrees of ripeness.

## 4. Conclusions

This study evaluated the effects of using four non-*Saccharomyces* strains in fruit winemaking, with special attention to the action of the *Metschnikowia pulcherrima* clade. The composition of the fruit wine was dependent on the type of fruit matrix and the yeast strains used as co-cultures. The chemical changes in the tested fruit wines were compared to wines fermented with *S. cerevisiae* as the sole starter. Non-*Saccharomyces* can modulate the chemical nature of fruit wines. Hydrolytic enzymes such as proteases, glucanases, β-glucosidase, lipases, and esterases make *M. pulcherrima* a very interesting fermentation partner for *S. cerevisiae* in apple wine. The obtained apple wines were characterized by lower ethanol content, high glycerol levels, and higher amounts of compounds creating the wine volatilome. However, the apple/chokeberry matrix was more difficult for winemaking, and the effects of other non-*Saccharomyces* yeasts on the chemical character of the wine were not as pronounced. The use of *M. pulcherrima* as a starter in mixed fermentations with *S. cerevisiae* could be of great interest in modern fruit enology. However, more research is needed on the impact of different types of fruit matrices and other types of co-inoculation on the characteristics of wine. Large-scale research using industrial-quality fruits harvested from different climate regions is particularly necessary. It would also be interesting to study the bioprotection potential of *Metschnikowia* spp. and their mechanisms of action during fruit wine fermentation on an industrial scale.

**Author Contributions:** Conceptualization, D.K.; methodology, D.K., K.P.-P., U.D.-K. and E.P.; validation, K.P.-P. and U.D.-K.; investigation, E.P.; data curation, H.A.; writing—original draft preparation, H.A., U.D.-K. and K.P.-P.; writing—review and editing, D.K.; visualization, D.K.; supervision, D.K. All authors have read and agreed to the published version of the manuscript.

**Funding:** This research received no external funding.

**Institutional Review Board Statement:** Not applicable.

**Informed Consent Statement:** Not applicable.

**Data Availability Statement:** Not applicable.

**Conflicts of Interest:** The authors declare no conflict of interest.

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
