# Peer review of "Exploring Use of the Metschnikowia pulcherrima Clade to Improve Properties of Fruit Wines"

_fermentation, doi:10.3390/fermentation8060247_

Round 1

Reviewer 1 Report

This manuscript is overall well-written dealing with the aroma composition of different fruit wines using different yeast. There are some suspect findings that needs to be dealt with before this can be accepted. 

Major issues: S. cerevisiae should not be able to utilize arabinose and most likely also not cellobiose as claimed by table 3. There has been many engineering attempts to enable S. cerevisiae to utilize these carbon sources. There should be an explanation for this. There are some "wild Saccharomyces" that could have B-glucosidase activity but arabinose I find hard to accept.

W. anomalus has more fructose (table 5) after fermentation than at the start (table 2)? How is this possible? 

Small issues. The title: even though Metschnikowia is the main feature of the paper, basically all the fermentations and subsequent analysis were also done using other yeast and it is not reflected by the title.

Introduction: it is a bit of unnecessay part mentioning plums and cherrys

There are literature which contradicts Reference 15 https://doi.org/10.1016/j.fm.2020.103670

Table 3 and in the text ...sucrose is better to use than saccharose

Table 6: take care for the significant values

All the table where S. cerevisiae is the control: just place an "a" next to each value of S.cerevisiae to make it clear.

Author Response

We would like to take this opportunity to express sincere thanks to Reviewer 1 who identified areas of the manuscript in need of correction and modification.

Authors

Reviewer 2 Report

The manuscript describes the effects of four non-Saccharomyces strains in fruit winemaking, with special attention to the action of the Metschnikowia pulcherrima 
Hydrolytic enzymes (proteases, glucanases, β-glucosidase, lipases, and esterases) are key enzymes for successful fermentation processes in apple wine. 
The use of M. pulcherrima as a starter in mixed fermentations with S. cerevisiae is an interesting approach that could be of interest in fruit enology. 
I think this manuscript will be useful for readers and I recommend its publication.

Author Response

Thank you very much for your positive feedback on our manuscript. Thank you very much for taking the time to share your experiences with us.

Reviewer 3 Report

The authors describe in the paper: Exploring use of the Metschnikowia pulcherrimaclade to improve properties of fruit wines", the influence of different non-saccaromycetes on the composition of apple and apple/ chokeberry wine. The work is of great interest and in principle very well written.

However, there is a lot of confusion regarding the variant description. Please explain at the beginning of Material and Methods which variants were performed and with which number of replicates.

Furthermore, it is not clear to me how the quantification of the aroma substances was done. Did you use analytical standards, or did you choose an internal standard and a semquantitative methodology? This definitely needs to be explained in more detail. Even if there is a citation here, information on quantification and validation should at least be given as supplementary data. The same applies to all HPLC methods!!! This is a big issue. Please give more information regarding the methods and the validation!!!!!!

The ANOVA requires a normal distribution. Normally, if there is no normal distribution and variance homogeneity, you should use a robust method (for example: Kruskal Wallis test).

Table 6: not acetals but aldehydes

Also, I have a problem with the Conclusio. We know that fermentations on a scale below 50 liters do not allow us to draw conclusions about the fermentation dynamics and aroma composition of wines on a large scale. Therefore, it should be noted that the results should be verified on a large scale.

There are a number of typing errors in the paper, so please check the English again. 

Considering all the needs I supose for major revision. 

In view of all these needs, a major revision is required. In particular, the methods are not described in detail. It is not enough to describe the analytical process, but analytical methods must be validated to confirm that the results are correct. If you use only semiquantitative methods, this would not be a problem for the Conclusio either, but then all tables would have to be changed and always the concentration mg/l Internal Standard would have to be given. 

Author Response

We would like to take this opportunity to express sincere thanks to Reviewer 3 who identified areas of the manuscript in need of correction and modification.

Thank you very much for taking the time to share your experiences with us.

Authors

Round 2

Reviewer 1 Report

The authors addressed my queries adequately

Reviewer 3 Report

Thanks for the changes